# Rapid Detection of Botulinum Neurotoxins—A Review

**DOI:** 10.3390/toxins11070418

**Published:** 2019-07-17

**Authors:** Robert J. Hobbs, Carol A. Thomas, Jennifer Halliwell, Christopher D. Gwenin

**Affiliations:** Applied Research in Chemistry and Health (ARCH) Research Group, School of Natural Sciences, Bangor University, Bangor, Gwynedd, Wales LL57 2UW, UK

**Keywords:** Botulinum Neurotoxin, Botulism, Rapid Detection, Sensitivity, PoC

## Abstract

A toxin is a poisonous substance produced within living cells or organisms. One of the most potent groups of toxins currently known are the Botulinum Neurotoxins (BoNTs). These are so deadly that as little as 62 ng could kill an average human; to put this into context that is approximately 200,000 × less than the weight of a grain of sand. The extreme toxicity of BoNTs leads to the need for methods of determining their concentration at very low levels of sensitivity. Currently the mouse bioassay is the most widely used detection method monitoring the activity of the toxin; however, this assay is not only lengthy, it also has both cost and ethical issues due to the use of live animals. This review focuses on detection methods both existing and emerging that remove the need for the use of animals and will look at three areas; speed of detection, sensitivity of detection and finally cost. The assays will have wide reaching interest, ranging from the pharmaceutical/clinical industry for production quality management or as a point of care sensor in suspected cases of botulism, the food industry as a quality control measure, to the military, detecting BoNT that has been potentially used as a bio warfare agent.

## 1. Introduction

Botulinum neurotoxins (BoNTs) are one of the most potent toxins known to man. With a median lethal dose (LD_50_) of 1–5 ng/kg [1], they are even more toxic than Sarin, 420 mg/kg [2], Ricin, 1.5 mg/kg [3], and Novichok, 220 ng/kg [4]. BoNTs are formed as proproteins with a single polypeptide chain of around 150 kDa by bacteria that originate from the *Clostridium genus* that are gram-positive, spore forming and are anaerobic, meaning that they occur in an environment that is free of oxygen. These are summarised in Table 1 [5].

*Clostridium Botulinum* can grow in the soil, spoiled food, in injuries that have broken the skin, or the human bowels, and can easily be grown in the laboratory [6]. Due to the high potency and lethality of BoNTs, their use as biological weapons is an ever-present possibility, and as such, they form one of the six category A agents. This is the list of agents that pose the highest risk for bioterrorism, as compiled by the US Centres for Disease Control and prevention (CDC) [7].

BoNTs also have important therapeutic and healing uses, with the toxins being utilised in treating conditions including, but not limited to: cervical dystonia, strabismus, blepharospasms and multiple sclerosis [6]. The use of botulinum neurotoxin serotype A (BoNT/A) is widely known for its application in the commercially available product Botox® in the cosmetics and beauty industry [8].

The toxin is activated through proteolytic cleavage into two parts; the first being a light chain peptide fragment (50 kDa), which is the zinc containing catalytic portion, and a heavy chain fragment which is roughly double the size (100 kDa), which comprises the receptor binding and central translocation domain. The two chains are linked primarily with a disulphide bond but also multiple interactions which are non-covalent in nature [9]. BoNTs can induce flaccid paralysis by attacking the SNARE (soluble N-ethylmaleimide-sensitivity (NSF) attachment protein receptor) proteins within neurons. BoNT internalisation begins with the toxin binding to two receptors situated on a neuron cell membrane, gangliosides and protein receptors, through its heavy chain [10]. The BoNT undergoes uptake into the neuron via endocytosis, where a change in pH within the endosome causes the toxin to change conformation allowing the light chain to exit into the intracellular fluid (ICF) [11]. Once within the ICF the toxin cleaves the proteins of the SNARE complex. This complex is comprised of synaptobrevin, SNAP-25 and syntaxin.

There are now eight different serotypes of the toxin labelled A to FA after the discovery of BoNT/FA (originally named type H until further investigation identified it as a chimeric toxin instead of a separate serotype) in 2016 [12,13]. There were initial concerns that this new toxin could not be neutralised by existing antibody products. Pellet et al. demonstrated in 2016 that existing type A antitoxins are effective [12,13]. Following on from the work of Pellet et al., another novel BoNT type was identified; it was discovered in a bivalent strain *C. botulinum* B strain and subsequently labelled BoNT/X [14]. Gene sequencing has identified a growing bank of subtypes for each toxin serotype. For instance the BoNT/A, BoNT/B and BoNT/F serotypes can be further classified into BoNT/A1 through to A8, BoNT/B1–B8 and BoNT/F1–F8, respectively. Additionally, BoNT/E can be separated into a larger subgroup covering BoNT/E1–12. Each serotype cleaves one of the SNARE proteins in a different site, as seen in Figure 1 and summarised in Table 2 [6,15].

The core SNARE complex consists of four α-helices; these come from the three proteins, synaptobrevin, syntaxin and SNAP-25 (which contributes two, abbreviated as Sn1 and Sn2) [16]. The toxin then moves to the SNARE proteins where it selectively cleaves one of these proteins at different scissile bonds (a covalent chemical bond that can be broken by an enzyme) depending on the toxin’s serotype. SNAP-25 is cleaved predominately by three serotypes A, C and E at residues 197–198, 198–199 and 180–181 respectively. Syntaxin is fragmented by serotype C between residues 253–254. Synaptobrevin (VAMP) undergoes cleavage via by serotypes B, D, F and G between amino acids 76–77, 59–60, 58–59 and 81–82. Importantly, once a molecule of toxin has cleaved, one molecule of the target protein is released, and the active site is regenerated to cleave further molecules of the target protein [17].

Infection by the toxin causes the disease botulism, which first presents with blurred vision, oral dehydration, difficulty speaking and laboured swallowing quickly followed by flaccid paralysis [18]. The most prevalent forms of the disease present as food-borne botulism, infant botulism and also wound botulism [19]. Treatment is through the administration of botulinum antitoxin and mechanical ventilation and should be provided as soon as possible as the antitoxin can only neutralise toxin molecules that are yet to be immobilised onto nerve endings [20].

## 2. Botulism: Types and Medical Application

### 2.1. Human Botulism

In humans, the disease is predominately caused by the toxin serotypes A, B, E and F and, as mentioned above, normally manifests in one of three naturally occurring forms: either food-borne, wound or infant botulism [18,21]. The BoNT undergoes uptake into the body via two main routes, the gastrointestinal tract or through membranes comprised mainly of mucous, such as those in the eye or respiratory tract. Once uptake is complete and the BoNT is internalised, it then proceeds to assimilate into the blood and lymphatic system. Upon completion the toxin then transverse to nerve endings, where it ultimately accumulates and impedes the release of neurotransmitters. Patients usually present with the medical difficulties described above and remain the same regardless of the serotype that is the root cause. This is a fairly rapid process with occurrences of flaccid paralysis present in as little as 12–48 h post toxin exposure [6]. Recovery can take anywhere from a matter of weeks to extended periods covering many months [22]. The recovery timescale is primarily dependent on the total concentration of BoNT ingested but the serotype involved in intoxication can also play a role, for example BoNT/A is generally observed to have potency levels higher than BoNT/B and BoNT/E [22].

### 2.2. Foodborne Botulism

Foodborne botulism is the form of human intoxication that is primarily observed and this happens when the *Clostridium botulinum* undergoes a period of growth leading to the production of toxins in food before it is consumed [22]. The bacterium is a gram-positive anaerobic bacterium; this means that for it to undergo growth, it must be in an environment that lacks oxygen. The growth of the bacteria and the subsequent formation of BoNT primarily occurs in products that have a reduced oxygen content, but also in those that have certain combinations of packing and storing temperatures as well as specific preservative factors [23]. Foods commonly involved are those that are prepared in the home and can include products like dried or salted meats, tinned produce and fermented fish [22]. Typically, these occurrences of intoxication are sporadic and limited to a household, with symptoms typically appearing within 12–72 h after toxin ingestion [22]. Occasionally, commercially prepared foods are involved [22]. The bacterium *C. botulinum* favours non-acidic environments (pH > 4.6), this means that BoNT does not get produced in foods that are acidic in nature [24]. A product with a low pH will also not have the ability to degrade any BoNT that may have formed prior to the onset of acidic conditions, e.g., earlier in the production process [24]. To hinder and ultimately avert bacterium growth and BoNT production, not only is pH manipulated, but various arrangements of parameters such as salt concentration and storage conditions are also utilised. [24]. Case fatality of foodborne botulism in developed countries is 5–10% [22]. In cases where there is an outbreak of BoNT, produce samples must be collected instantly, correctly, adequately stored and sent for analysis to quickly ascertain the origin of the contamination. This will assist in the prevention of any subsequent issues and in the remediation cause to bring closure to the outbreak and stop any ongoing issues from arising.

### 2.3. Infant Botulism

Infant botulism is defined as an illness that manifests in children who are less than one year old, with the majority of cases reported involving infants that are less than six months old. While the mechanism is similar to that of foodborne incidents of the disease, the primary difference is that it is the ingestion of BoNT in produce responsible for foodborne botulism whereas in cases classified as infant botulism it is ingestion of *C. botulinum* spores, which in turn undergo growth to produce the bacteria which then release BoNT when accumulating in the digestive system of the young child [25]. Generally, in those aged over 6 months, this production of the bacterium is inhibited by the body’s normal immune responses, which have had a chance to progress in older children and adults [25]. Symptoms of botulism in infants differs from those generally experience with the illness with these including; decreased neck muscle control, myasthenia, a cry that is different in tone or pitch to a child’s normal response, irregular or hard bowel movements and even appetite decrease [25]. While there are a multitude of potential causes of infection, one that is frequently linked to several infant botulism cases is that of honey that has been tainted with the spores [25]. The result of this link has seen advice provided to both parents and guardians to avoid children under one year of age from consuming honey and honey containing products. 

### 2.4. Wound Botulism

The frequency with which wound botulism is observed is very sporadic and it presents when the bacterium spores become exposed and embed into a break in the skin, they must penetrate deep enough in the wound in order to undergo growth in anaerobic conditions [26]. Although people suffering from wound botulism will suffer in the same manner as those with foodborne botulism, the resulting symptoms normally take an increased period of time to manifest, due to the slower progression of the spores and bacterium, typically the onset of symptoms can take as long as 10–14 days to present. Wound botulism is commonly coupled with occurrences of complex infections and can often be attributed to more than a single BoNT serotype [26]. One of the more common causes for this manifestation of the botulism disease is the injection of drugs and substance misuse, frequently from contaminated drugs such as black tar heroin [26].

### 2.5. Inhalation Botulism

The route of exposure required for inhalation botulism to manifest is not a normal method of exposure and is generally linked with either unintentional release event, or perhaps more alarmingly, via intentional release as a result of a bioterrorism event resulting in the discharge of toxins as a suspension of fine particulates [23]. The only reported cases of inhalation botulism were recorded as a result of the exposure of three laboratory technicians to the toxin in a research facility in Germany in 1962. This was as a result of unintentional exposure that occurred during an autopsy of some animal test subjects in the research facility; these animals had been prior exposed to BoNT serotype A, this exposure resulted in the technicians exhibiting botulism symptoms [27]. The incubation period generally ranges between 24 and 36 h, and up to 5–6 days post exposure. Clinical symptoms are similar to those seen with foodborne botulism except for the absence of digestive system-based symptoms [28]. The LD_50_ via the inhalation route for humans has been estimated at 700 µg to 900 mg assuming a bodyweight of 70 kg [7]. Laboratory confirmation of inhalation botulism is difficult because the toxin is not typically distinguishable in the serum or stool whereas it is in foodborne cases [28].

### 2.6. Other Types of Intoxication

Ingestion of the pre-formed toxin from a water source could occur, but standard water treatment processes such as boiling and disinfection with 0.1% chlorine containing bleach solution, such as sodium hypochlorite, would normally result in the destruction of the toxin [29]. The likelihood of toxin exposure proceeding via this route is considered to be minimal [29]. Where the origin of the botulism exposure in adult cases is not attributed to either foodborne or wound exposure it can prove difficult to identify and remains undetermined, it is likely to have occurred where antibiotic therapy or a surgical procedure has altered the normal intestinal flora [23]. Iatrogenic botulism is another manifestation of the disease and is typically reported as a result of the use of BoNT in medical or cosmetic products, e.g., Botox® [30,31].

### 2.7. Botulism in Animals

Serotypes C and D are more commonly responsible for disease in most animals. In Europe, botulism has become a disease of interest due to its emergence in the commercial chicken sector in both egg and meat production [32]. Birds are typically exposed to BoNT through the ingestion of invertebrates, after which BoNT levels in the blood reach levels that induce the signs of paralysis [32] In Brazil the disease is widespread in cattle [33]. Dogs can become affected through eating rotten food or infected carcases [34]. Horses, however, are most commonly affected by serotype B, which in adults is acquired through the ingestion of preformed toxins in the feed (in a similar way to human foodborne botulism) [35]. The disease course is related to the overall toxin exposure and frequently results in fatal consequences without prompt treatment with specific antitoxin [35].

### 2.8. Medical Applications of Botulinum Neurotoxin

BoNT works via suppression of transmitter release from endings of motor neurons, resulting in flaccid paralysis [36]. Injected BoNT inhibits release of the neurotransmitter acetylcholine, therefore preventing contraction of muscle cells [8,37]. BoNT/A is currently used to treat over 20 different medical conditions, including: Blepharospasm (spasm of the eyelids), cervical dystonia (neck and shoulder spasm), chronic migraine, excessive sweating, strabismus (eye muscle disorders), post-stroke upper limb spasticity, urinary incontinence and hemi facial spasm [8]. Perhaps the most well-known use is as Botox® injections for cosmetic purposes, such as the treatment of glabellar lines (frown lines) and crow’s feet. In fact, around 50% of all medical production of BoNT is currently utilised for aesthetic medical products [36]. Injections are generally well tolerated with few side effects. Around 1% of people develop antibodies that make subsequent treatments ineffective [38]. Due to the fact that the toxin is a biological product which intrinsically leads to variability between production batches, each set has to be tested for safety, potency and stability before it can be used on humans [39]. The testing must be conducted at multiple steps in the manufacturing timeline, this is a mandatory requirement to be able to meet regulatory protocols, leading to marketing consent being approved [40].

## 3. Methods of Detection

The quick onset of the disease, the extreme toxicity of BoNTs and the absence of treatments to reverse paralysis [6] means that a quick BoNT detection method is required that is both sensitive and specific. It also needs to be as wide reaching as possible and be compatible with food and environmental samples. The ability to reliably detect BoNT has applications outside of clinical diagnostics [6,41,42]. As previously stated, BoNT represents a significant bioterrorism threat, and if an attack event were to happen, then a detection method that combines speed, sensitivity, ease of use and the ability to be used in various potentially harsh environments would be required to evaluate the magnitude of any contamination [7]. The growing medical use of BoNT means that its definitive detection in the production process and research laboratories is of equal importance [6]. This review will provide an unbiased comparison of our own methods, along with other promising detection methods and compare them against the current gold standard, the mouse bioassay. Most developments towards animal replacement methods have been focused on the potency testing of pharmaceutical BoNT preparations. This work has led to two FDA-approved methodologies; both are cell-based potency assays, firstly as a result of the work in 2012 by Fernández-Salas et al. [43,44] and secondly by the Merz Pharma Group in 2015 [43,45]. There have also been other cell-based assays (CBA) achieving levels of approval with Allergan (Canada, USA and Swiss approval 2011, EU approval 2013) and Ipsen (EU and Swiss approval 2018) providing CBA detection methods [46]. The ability to quantify active levels of toxin, like the MBA can, is of high importance, predominantly for pharmaceutical purposes. This will be identified in this review, along with comments on the detection methods ability to detect either multiple serotypes or within complex sample matrices, which are characteristics more prevalent with diagnostics and food testing for BoNT. Finally, the review will compare the methods against three criteria; speed of detection, sensitivity of detection and finally cost all of which are more important in food and diagnostic testing. Despite the advances made, it is estimated that in the EU around 400,000 animals per year are still being subjected to batch testing [46]. With this number contributing to a worldwide figure of approximately 600,000, of which 70,000 animals are tested in the UK alone, highlighting the need to further explore replacement detection methods [42]. 

### 3.1. Mouse Bioassay

The mouse bioassay (MBA) remains the most widely used test to confirm levels of active BoNTs [23]. The test works through injections within or through the peritoneum of samples that are suspected to contain the toxin. The mice then undergo observation for signs and symptoms that the disease is present, which include pilo-erection, wasp-waists, hind limb paralysis, dyspnoea and ultimately death by respiratory paralysis [47]. This typically occurs within 48 h post injection [6]. It is essential to determine the amount of BoNT in a sample, and this is done by quantifying both the maximum sample dilution that results in fatality in the mice, and also the minimum dilution that does not result in mouse death. This part of testing may require repeating if the dilution which does not kill is not found first time [48]. The serotype of the toxin is determined through administration of serotype specific antitoxins prior to injection with the sample. The mice are observed for signs of botulism for a further time period of 48 h to confirm that the specific antitoxin for a serotype in the sample is protective. Therefore, at least 4–6 days are needed to carry out this assay. This assay is considered to be very sensitive, detecting down to 10 pg/mL, and it is also able to detect functionally active toxin, which is in direct contrast with a vast number of other immunological methods [6,39,41]. It does, however, have several disadvantages, including the cost and ethical issues of live animal research, as well as the amount of time taken to conduct the assay [49,50]. 

### 3.2. Enzyme-Linked Immunosorbent Assay

Enzyme-linked immunosorbent assay (ELISA) is a biochemical procedure that utilises antibody conjugated enzymes to detect the presence of a specific antigen [51]. ELISA has wide-ranging applications and is regularly used to diagnose a variety of diseases in medicine and as a quality control check in many industries, including to test for cross-contamination in food production [52,53]. There are three common set ups: direct, indirect and sandwich (capture), as shown in Figure 2.

The different methods vary with respect to how the antigen is bound and the number of antibodies used. For direct and indirect ELISA, the sample to be tested is first applied to the microtiter plate, allowing any antigens present to bind before blocking solution, commonly bovine serum albumin (BSA) or casein, is added in order to prevent any non-specific binding of antibodies [54]. Next, the primary antibody that is specific to the antigen is added and allowed to bind. In the case of direct ELISA, this antibody is conjugated to an enzyme which on addition of its substrate produces a measurable colour change signal [55]. With indirect ELISA, a secondary antibody which contains the enzyme conjugate is used to produce the response. This antibody binds to the Fc portion of the primary antibody so does not need to be specific for the antigen [56]. These antibodies are readily commercially available and remove the need to produce antibodies specific for the antigen with the enzyme conjugate, which is a time-consuming and costly process. In sandwich ELISA, a capture antibody (c-Ab) is used that specifically binds the antigen before the surface is blocked and the antigen, primary antibody and secondary antibody conjugate layers are built up as in indirect ELISA [57]. This allows for a more sensitive result, as there are two specific antibodies, and as such, this is the most commonly used for BoNT ELISAs. Using serotype-specific antibodies allows for the determination of which serotypes make up a toxin sample, making it possible to administer the correct antitoxin quickly which is essential for successful treatment of botulism [58]. The detection limit for ELISA ranges from 2 pg/mL to 2 ng/mL, with typical assay lengths of 5–6 h [6,41].

The detection of BoNT complexes in immunological assays is hampered by the presence of neurotoxin-associated proteins (NAPs). It is this NAP complex that blocks antigenic sites of the toxin, making them unavailable for binding to antibodies, meaning many assays are developed using highly purified BoNT samples, which is not advantageous for use in diagnostic and food-testing [59]. This has led to several groups focusing on development, characterisation and screening of new antibodies that recognise free epitopes on the toxin overcoming the problem of NAPs [60,61,62,63,64,65].

### 3.3. Immuno Chromatography Assays—Lateral Flow/Column Flow

Lateral flow assays (LFAs) are hand-held assays, most well known for their use as pregnancy tests. For the detection of BoNT, these assays utilise antibodies conjugated to colloidal gold to yield a colour response in the presence of the toxin [66]. The sample is added on a sample pad where any toxin present attaches to antibodies on the colloidal gold. They then migrate up the nitrocellulose strip to the capture antibodies where, if positive, they bind, producing a colour response. Excess conjugate then travels further to the control antibodies, where it binds and produces the second colour response showing that the sample has migrated correctly, as shown in Figure 3.

LFAs have several advantages when compared to other detection methods: they are very low in cost, self-contained, require no sophisticated equipment or expert analysis and have a very rapid analysis response time of around 15 minutes. The combination of advantages makes them a prime candidate for field use. Nonetheless, LFAs do have poor levels of sensitivity compared with other detection assays such as ELISA and similar immunological techniques. Normal detection limit ranges when using a detection antibody conjugated with gold nanoparticles vary between 5 and 50 ng/mL [68]. Increased sensitivity has been seen when the reporter is replaced with horseradish peroxide (HRP) or silver enhancement. The latter increases the sensitivity to 50 pg/mL [67]. The other main drawback is that due to the use of antibodies, as in the ELISA method platform, the LFAs are unable to differentiate between active and denatured toxin [66,69]. A key requirement of MBA replacements in relation to their usage in the pharmaceutical production sectors is the ability to quantify active toxin levels, so the inability to do this presents a significant hurdle. In 2017 Liu et al. achieved an increase in sensitivity for a LFA system, their system was capable of detecting as low as 20pg/mL for BoNT/A using a very small sample size (1 μL) [70]. To achieve the improvement, Liu et al. reconfigured gold nanoparticle-based lateral flow strips with specific substrate peptides that integrate endopeptidase activity to the assay [70]. It is proposed that the technology could be extended to other BoNT serotypes by designing and encompassing more specific substrate peptides that correspond with the additional serotypes. This would be required for the method to progress for uptake into the diagnostic and food testing sectors [70]. There are, however, two major drawbacks for this methodology, which are that the speed of analysis for the improved system decreased to around 12 h and there was also an increase in cost due to the integration of the endopeptidase activity assay [70].

### 3.4. Immuno-PCR/Liposome-PCR

Immuno-polymerase chain reaction (Immuno-PCR) is an ELISA-type immunological test that uses polymerase chain reaction (PCR) to increase amplification of the ELISA signal. The detection methodology relies on forming complexes of antigen and antibody, in the case of Immuno-PCR, its differentiating feature is that it utilises the formation of an antigen-antibody complex, which is then bound to known DNA molecules instead of the normally used enzyme format. Once the binding has occurred to form the complex, the amplification of the DNA fragments which are bound to the BoNT unique antibody is easily implemented using traditional or real-time quantitative PCR (qPCR) [6,41]. Immuno-PCR has been found to detect botulinum neurotoxin serotype A to sensitivity levels similar to those associated with the MBA. It possesses the ability to determine active toxin levels, which is a major boost to its potential as an MBA replacement in the pharmaceutical production sector [71]. The use of streptavidin as a bridging molecule to link biotinylated modified DNA tags and antibodies has seen a reported sensitivity of 1 pg/mL in relation to serotype A [72].

In the liposome-PCR test, around 60 copies of reported DNA are encased within a liposome. It has also had its outer surface labelled with specific molecule, for BoNT binding this molecule is typically trisialoganglioside (GT1b) [73,74]. The modified surface of these laden lipid-based vesicles are able to conjugate with a complex made up of a capture antibody and the toxin of interest, this is then followed by disruption to the vesicles, and qPCR of released DNA. Using this method BoNT/A was detected in purified water at levels down to 20 pg/mL [73], causing this assay to have a sensitivity detection level around 100,000 times lower than the mouse bioassay. One of the main drawbacks of this particular detection method is that it has not been used and validated in other environments such as clinical or food samples, which would prevent it from having extensive application in the diagnostic and food-testing sectors. Both Immuno-PCR and L-PCR assays take around 9 h to run [35].

### 3.5. Enzyme-Linked Immunosorbent Assay on a Chip (EOC)

The combination of two detection methods is the basis of cross flow immunochromatography. The amalgamation of ELISA, lateral flow assay and utilisation of biosensor automation led to the progression of an EOC system for the detection of botulinum neurotoxin serotype A [75,76]. The basic design of the EOC system for BoNT/A detection is made up of a top polycarbonate moulded plate with two traversing channels recessed into the underneath, along with a horizontal flow adsorption pad. An immuno-strip containing two serotype A-specific heavy chain antibodies is then contained between the top and bottom polycarbonate casings using a UV sensitive adhesive [75]. The two antibodies were used to fulfil two specific roles in the EOC device; one was immobilised onto the membrane to be used as a capture antibody, the second was conjugated with HRP and utilised as a detection antibody [75]. Upon applying a sample containing the desired BoNT/A analyte, the EOC device produced a colour signal that was linearly correlative to the concentration of BoNT/A available [75]. The strength of an observed colour signal was able to be quantified using a detector system equipped with a high-quality digital imaging. The EOC system was able to analyse BoNT/A with an LoD of 2 ng/mL with an analysis time of less than 30 minutes [76]. 

### 3.6. Biosensors

Biosensor technologies embody a wide and diverse range of BoNT detection methods, usually, the base platforms used consist of: surface plasmon resonance (SPR), refractometer, fluorescence and chemical luminescence. Typically, evanescence wave technology is used for fluorescence-based biosensors. Molecules that are labelled with fluorophores and bound to a surface become excited upon exposure to evanescent fields, resulting in the production of a signal. Often the assay used is an immunological sandwich assay, consisting of two antibodies for immobilisation and quantification along with the analyte of interest, all of which subsequently locate on the sensor surface. Utilising this system, the sensitivities of sensors tested ranged from as little as 150 pg/mL [77] to 200 ng/mL [78] for botulinum neurotoxin serotypes E and B, respectively. These results were generally obtained from analyte samples in simplified buffer systems, which would limit wide-scale adoption in the diagnostic and food testing fields. However, it has been reported that serotype A detection can achieve an LoD of approximately 50 ng/mL even in more complex matrices such as food samples [79].

The use of aptamers, which can be defined as oligonucleotide fragments that can achieve protein-specific binding, has advantages over the use of antibodies, including an easier screening method, increased stability, and their sustainable usage [80]. This technique combines an electrochemical approach integrating enzymatic amplification with an aptamer probe, typically around 70–80 bp in length. This will then undergo a structural change to its standard conformation, this is facilitated by the binding of toxin present in the sample [80]. The aptamer was bi-labelled with both biotin and fluorescein. Post-binding to toxin the aptamers structural change introduces a conformational opening allowing the fluorescent reporter tag in which lead to the generation of an electrical response. Consequently, only a specific toxin can generate an amplified current [80]. The LoD for this aptamer-based system was quantified as 40 pg/mL (BoNT/A) which is similar to sensitivities seen in traditional ELISA, but is unable to determine levels of active toxin which is barrier in the way of potential application in pharmaceutical production testing [80].

Generally, biosensor-based platforms take >20 minutes to complete, with multiple analytes detectable simultaneously, depending on the sensor design, this would be of particular interest in development of the system for its use in the diagnostic and food testing sectors. The speed with which results can be obtained make biosensors one of the most rapid available platforms around. The desired rapid analysis, however, comes at a cost, as is seen with LFA devices, leading to limited analytical sensitivity. However, a review of recent progress in the field of biosensors for their use in detecting toxins concluded that there have been great advancements based on transducer parameters as well as bio-recognition elements [76]. It was felt that the progressive systems could make the transition from laboratory to commercial applications within the next few years [76]. Evidence of increased sensitivity is demonstrated by the recently developed Newton Photonics SPR biosensor, which has a LoD quantified at 6.76 pg/mL (BoNT/A light chain), allowing for active toxin quantification, which is advantageous for its adoption in the pharmaceutical production sector. The SPR method has a detection time of less than 20 minutes [81].

### 3.7. Fluorescent Resonance Energy Transfer Assay (FRET) 

An observable variation in fluorescence of a substrate upon fragmentation is commonly utilised in the detection of enzyme endopeptidase activity. In FRET assays, Figure 4, an oligopeptide which mimics a natural substrate is used which carries two tags either side of the cleavage site. One is known as the fluorescent quencher, the other is the fluorescent donor [82]. FRET is detected only when the two tags are close to each other, allowing transfer of fluorescent energy from donor to quencher. 

When the substrate is fragmented after it undergoes cleaving by the toxin, the tags are separated, leading to the energy transfer being inhibited [41]. The decrease in FRET is linearly correlative to the toxin concentration. The LoD of FRET-based platforms is dependent upon the substrate incorporated into the assay but is normally within the region of 40 ng–60 pg/mL [6]. 

The average speed of analysis via FRET is around 3 h [6,82,83], although Guo et al. reported an improved analysis time of 2 h, although this was with BoNT/B [82]. Combining the assay with an immunoseparation step enhances the sensitivity to around 1 fg/mL, approximately 100,000 times more sensitive than the MBA [83]. In this assay, the toxin is first captured at the end of the immunoseparation step, whereby the toxin is isolated using toxin specific antibodies bound to microbeads. These beads are then re-suspended in endopeptidase reaction buffer, which also contains synthetic FRET substrate to initiate cleaving of the toxin, allowing for determination of active toxin levels, which is required for MBA replacement in the pharmaceutical production field [83].

### 3.8. Flow Cytometry

Flow cytometry equipment can be used to analyse and quantify toxins using fluorescence immunoassays [6,41,84]. A multiplexed assay to test both BoNT/A and B alongside ricin, abrin and Staphylococcus enterotoxin B in a diverse range of food matrices utilising the application of fluorescent magnetic beads, detected toxin levels of 21 and 73 pg/mL, respectively [85]. The assay was enhanced by using an automatic fluidic system to handle the sample. Beads alongside capture antibody are contained in a flow chamber, where the toxin is first captured from the test matrix, it then undergoes washing and subsequently capture via antibodies before, finally, FC investigation. The sensitivity limit was determined as 50 pg/mL in relation to the heavy chain fragment of BoNT/A, allowing quantification of active toxin levels, with an analysis time of 4 h [86]. This characteristic of the methodology is a key component of a replacement methods ability to be used in the pharmaceutical sector.

Flow cytometry assays have several advantages over ELISA. They are easier to automate, and therefore it is easier to detect multiple toxins and/or botulinum serotypes in a single sample, which is key to the widespread adoption of a detection technique in the diagnostic and food testing fields. Other advantages of bead immobilisation of toxin compared to on a microtiter plate, including improved capture kinetics as well as improved analyte concentration [85,86]. This detection method has been tested in other environments and has been shown to be appropriate for BoNT detection in an extensive variety of produce matrices [85]. The principle drawback of flow cytometry is that it requires instrumentation that is significantly more expensive than that required for ELISA or the MBA [6,86].

### 3.9. Fluorescence Endopeptidase Assay

Fluorescence endopeptidase assays have been researched against several BoNT serotypes. In one particular example, peptides that were labelled with a fluorescent tag were formulated to be serotype specific and immobilised on a solid substrate. The BoNT would initiate a specific cleavage, mediating the release of a fragment of the tagged peptide into the solution surrounding the substrate. Spatial separation allowed recognition of the different serotypes A, B, E and F to a LoD of 2 ng/mL, the ability for a BoNT detection method to recognise multiple serotypes is of particular interest in both the diagnostic and food testing sectors [87]. This assay has been further developed into a semi-automated microfluidic format, utilising the same cleavage of a fluorescence-tagged peptide [88]. The main difference is that a toxin sample is added to the microfluidic device through an input entry point, allowing for an increased chance of successful cleavage of the fluorescent tagged peptide fragment from the solid substrate. The tagged fragments then pass through the device via a microchannel to the exit point, which aides in the evaporation of the solution, allowing the concentration of tagged fragments present to be increased. These modifications allowed the semi-automated device format to increase fluorescent signal amplification by 300%. The first generation of this device utilised silica beads as the solid substrate, but this allowed for an increased variation due to poor consistency related to bead loading, this ultimately affected the overall sensitivity of the device [89]. The issue was addressed by changing the solid substrate setup to that of a gold surface on which the fluorescence-tagged peptide could form a self-assembled monolayer [88]. The increase in sensitivity allowed BoNT/A to be detected down to 3 pg/mL with active toxin distinguishable, which is relevant for pharmaceutical adoption of a methodology. This was, however, for a purified sample, not a concern for pharma usage; nonetheless, when investigated in a more complex matrix, another key requirement of the diagnostic and food testing sectors, the LoD was drastically lowered to around 500 ng/mL with an analysis time of 3 h [88]. The decrease in sensitivity suggests the major drawback for this type of device moving forward is its inability to be utilised across a wide range of sample matrices, which would limit the usage uptake of a prospective commercial device and dramatically reduce its potential market.

### 3.10. Centrifugal Microfluidic Technology

The centrifugal microfluidic immunoassay platform offered by SpinDx is suitable for both bench and field detection of BoNTs [90]. The sample is mixed with a detection solution which is comprised of i) capture beads that have been surface functionalised with BoNT specific antibodies, ii) fluorescence-tagged detection antibodies, these become immobilised to the functionalised beads upon exposure to the equivalent antigen. This is incubated at room temperature, although a more recent modification to the device allows for changes to temperature due to the incorporation of a heating element. The mixture is then applied on a preloaded density medium, which is located within a channel present in the disc-based system. When this is centrifuged, the micro-particles diffuse across the density medium, resulting in a pellet at the extremity of the furrow. Fluorescence is measured, allowing quantification of the level of BoNT present, with the latest modification allowing the ability for the system to distinguish active toxin [90]. The whole assay takes less than 30 minutes and is as simple as loading a sample (which can be of a wide range of complex matrices) onto the disc and then placing it in the reader and commencing the testing. It also requires no additional sample preparation and can use a sample size of 2 µL. The complex sample matrices and minimal sample preparation lend themselves very well to the adoption of this methodology for diagnostic and food testing. This is, however, countered by the technology needing to be laboratory-based. This incurs high costs and requires trained personnel to interpret the results; this is also coupled with high equipment costs. In direct evaluation with the MBA, SpinDx was found to have a sensitivity of 90 ng/mL, 100-fold more sensitive than the MBA with the aforementioned active toxin detection a key component of MBA replacement in the pharma sector [90]. 

### 3.11. Electrochemiluminescence Immunoassay

Electrochemiluminescence (ECL) immunological assays depend on the similar reactions that are prevalent in the bulk of immunoassays. The main differing factor is the use of a electrochemiluminescence tag couple to the antibody responsible for generating signal, this modification allows the emission of light in the presence of a voltage [91]. ECL platforms are usually performed on c-Ab functionalised magnetic beads. Once binding of the toxin and reporter antibody has taken place, the beads undergo magnetic direction terminating in the vicinity of an electrode, which is where the ECL process proceeds [6,92]. Increases in sensitivity can come from multiple origins: (i) improved luminescent signal:noise ratio; (ii) bead functionalisation facilitates heavier antibody packing; (iii) improved kinetics of Ab-Ag interactions as a result of free flowing substrate; (iv) amplification of the assay results due to the quantity of beads that are held by the magnetic field facilitating an enhanced sample volume interacting with the beads [6,92].

In an early iteration of an ECL assay BoNT/A was found to have a LoD of 5 pg/mL and able to determine levels of active toxin which is of interest for method development in the pharmaceutical sector [92]. However, the sensitivity increase when evaluated against ELISA is minimal, as well as the diversity of matrices tested being more limited for ECL; this presents a disadvantage moving forward with this assay in diagnostic and food testing. The instrumentation required for ECL analysis is more specialist and more difficult to use for other purposes [92,93].

### 3.12. Cyclic Voltammetry and Electrochemical Impedance Spectroscopy

Electrochemical techniques are gaining traction for their use in sensor platform development. The main advantage being that they allow for a high sensitivity to be achieved in relation to surface interactions and changes [94]. Some of the areas in which these methodologies are being utilised include sensors for various biomarkers [95,96,97] and toxins [98], along with pathogens [99,100]; this is not only due to the high levels of sensitivity, but also because these techniques yield rapid detection times compared to the mouse bio-assay [101]. Both cyclic voltammetry (CV) and electrochemical impedance spectroscopy (EIS) have been utilised in the development of BoNT sensors [94,101]. Briefly, in CV a charge can be measured that relates to the number of species on a working electrode that are able to be oxidised, as well as the number of electrons involved in the oxidation [102]. In terms of a detection method, it is the variations in charge observed when there are modifications to the self-assembled monolayers formed on a working electrode surface that are of interest (Figure 5), as upon addition of an analyte, a binding event would see a change in the charge generated. 

When current moves through a circuit, consisting of a series of capacitors, inductors and resistors, the measure of the complex resistance that arises is called impedance. Electrochemical impedance spectroscopy uses redox probes including the ferri/ferrocyanide couple [Fe(CN)_6_^3−/4−^] or the hexaammineruthenium III/II couple [Ru(NH_3_)_6_^3+/2+^] and consists of measuring the ability of the ions in these redox couples to be both oxidised and reduced around the working electrode [103]. If a working electrode such as an Au (111) slide is not modified and the surface not blocked in any way, then the redox couples can easily undergo the redox reactions. If the surface of the electrode has been modified, for example, with a protein monolayer, then the ions in the redox couple are impeded from undergoing the oxidation and reduction reactions. This leads to an upturn in the charge transfer resistance of the circuit (Rct) (Figure 6) [104]. 

For BoNT detection, CV was used, exploiting the interaction between toxin and the SNAP-25 protein at the surface of an Au (111) electrode. The SNAP-25 protein lends itself to this setup, as it has four cysteine residues which form Au-S bonds to the surface of the gold electrode creating a self-assembled monolayer [94]. Botulinum neurotoxin is able to cleave 9 amino acids from the immobilised SNAP-25 facilitates a significant change at the surface which leads to a decrease in the anodic peak charge observed (Figure 5) [94]. The difference in charge before and after incubation with the BoNT sample is then calculated with a correlation between concentration of toxin and drop in charge observed, the higher the concentration of BoNT the larger the decrease in charge. The CV assays were able to quantify active BoNT as low as 250 pg/mL, and although this level of sensitivity is less than that observed in the MBA (10 pg/mL), it is on a similar level to the ELISA methods used for BoNT detection (~200 pg/mL) [94]. Active toxin detection is advantageous for the development of the method to be used in the pharmaceutical sector [94]. The main advantage of this assay is the speed of the assay, with the BoNT sample needing just 10 min incubation with the SNAP-25 modified electrode and the measurement taking another 1–2 min. This is significantly quicker than the MBA, and a lot of other BoNT detection methods [94]. It also has a relatively low cost, especially compared to the MBA, with each CV test costing ~£20, with the bulk of the cost coming from the gold electrode, which could be decreased if the method moved forward commercially.

An electrochemical impedance spectroscopy detection method was developed utilising the ferri/ferrocyanide redox probe to measure the impedance that occurred at the surface of a SNAP-25 and VAMP modified gold electrode [101]. Both proteins were used and tested due to their ability to detect different serotypes of BoNT, (Table 2). Upon incubation with the toxin the proteins immobilised on the electrode surface would become cleaved leading to the loss of a small fragment, ~9 amino acids (Figure 6). This leads to a measurable decrease in the Rct, allowing for the quantification of the concentration of BoNT present [101]. 

The EIS detection method was able to distinguish the presence of active BoNT to a sensitivity level of 25 fg/mL with slight variation between serotypes [101]. The ability to distinguish active toxin is a key component of MBA replacement assays, as it is the primary objective required in the pharmaceutical industry. The multiple toxin type testing is extremely advantageous for adoption in the diagnostic and food testing fields, although further study into a wider range of sample matrices would be required. As with the CV detection method the EIS method is not as sensitive as the MBA; however, the assay is still considerably quicker, with measurements taking around 35 minutes after introduction of the toxin [101]. The cost of the EIS is around £24, which is comparable to the CV detection method, with commercial uptake of the assay leading to a further decrease in potential costs.

### 3.13. Immuno-Detection of Cleavage Product

Immunological detection of BoNT cleavage products can be used in methodologies for the quantification of toxin and antitoxin potency. Classically, a synthesised peptide substrate, obtained from one of the SNARE protein, is exposed to BoNT. The products that result as a consequence of the cleaving are quantified with antibodies that are specific to the fragments [6,105]. Detecting cleavage products in relation to botulinum neurotoxins is both specific and quantitative and is consequently applicable for determining the potency of pharmaceutical used BoNTs [105]. Yadirgi et al. reported the introduction and improvement of a simplified direct ELISA platform for detecting BoNT/A activity in neurons with have been obtained from stem cells that come from pre-implementation stage embryos in mice [105]. The resulting product is caught using a specific neoepitope antibody, raised against a peptide that is equivalent to the 190-197 amino acid sequence of SNAP-25. The type of antibody used is selected due to its ability to only distinguish fragmented SNAP-25 and not the intact protein. The captured product is then detected with a two-site binding approach facilitated by dual polyclonal detection antibodies [105]. The limit of detection was stated as 440 ng/mL with an assay time of 6 h. The assay’s ability to determine active toxin concentrations is a major advantage in its relevance to replace MBA use in the pharmaceutical sector, subject to further testing and validation [105]. It is, however, limited in terms of its adoption as a diagnostic of food testing methodology due to the lack of serotypes tested and also the non-complex sample matrices used in the assay. 

### 3.14. Endopeptidase Mass Spectrometry 

Endopeptidase Mass Spectrometry (Endopep-MS) assays require the use of serotype-specific antibodies, see Figure 7 for diagrammatic overview. The antibodies are conjugated to magnetic beads and then added to a sample (a); the beads are removed and thoroughly washed before a substrate that imitates the toxins natural target is added (b). The solution is then incubated and the resulting mixture is analysed by matrix-assisted laser desorption/ionisation time-of-flight mass spectrometry (MALDI-TOF MS) [106]. The MS detects any whole substrate and any cleaved fragments that result from incubation with the active toxin (c), this is of particular interest if this methodology is to be utilised by the pharma companies for production testing [107].

The procedure takes around 4–8 h; while this is relatively lengthy, the ability to automate some of the steps in the detection method allow it to be adapted to process hundreds of separate samples in this time range. Rosen et al. have developed a multiplex platform to allow the assay to simultaneously detect BoNT/A, B and E [108]. This adaptation will further improve assay usability, particularly in the human diagnostic and food testing fields, by minimising the sample volume required and reducing the assay time [108]. The sensitivity achieved is good with a detectable range across 100 fg/mL to 1 ng/mL, depending on the sample matrix [109]. The wide range of sample matrices tested, including serum, stool, culture supernatants, and a diverse set of food samples again reinforces its potentially extensive application. Diagnostics and food testing require a varied range of sample matrices to be validated; this is not so much of a concern for the pharma sector, where simple buffers are the matrix of choice [109]. There are, however, problems with this assay; mass spectrometers are very expensive, and not many laboratories have access to one. It requires a trained technician to not only run the instrument, but also to interpret the data, which again narrows its wide scale adoption [108]. The length of time required to run a sample from preparation through to final analysis is also too long when compared with other techniques, which hinders the need for a rapid diagnostic or food testing detection method due to the quick progression of botulism [108].

### 3.15. UV/Visible Spectroscopy—Colorimetric Assay 

The basic principle behind the colorimetric assay is that there is simply a variation in colour that happens only in the presence of an analyte. Gold nanoparticles have been used to develop a diverse portfolio of biosensor platforms and diagnostic assays [110]. The tracking of colour change in relation to colloidal gold stems from the principle that certain wavelengths of light at the surface of the metal have the ability to excite the conduction electrons which in turn results in oscillation. It is this process that is referred to as localised surface plasmon resonance (LSPR) [111]. Gold colloids typically exhibit UV-visible spectra with distinctive absorption peak ranges of ~515–535 nm with respective nanoparticle diameters of 5–50 nm; this is due to their absorption of specific wavelengths of light due to LSPR [112]. The red colouration of the nanoparticles in colloidal gold is the result of them being negatively charged and its effect on the LSPR of the particles. The charge also results in the particles repelling each other remaining colloidal in nature. Upon addition of salt, the particles begin to aggregate together; this is due to the negative surface charges being masked and the particles no longer repelling each other [113]. Once aggregation of the particles occurs, they will begin to absorb longer wavelengths of light due to a decrease in inter-particle distance, which aids the coupling of the plasma modes. The increased wavelengths of light absorbed, typically in the region of 700 nm results in the colloidal solution undergoing a bathochromic shift, with the visible result observed being a change from red to blue [114]. This effect is able to be counterbalanced by modifying the surface of the AuNPs, an effective example is a covalently bonded protein layer, which bonds to the AuNP surface. This results in the particles being protected from the effects of salt addition preventing the masking of the stabilising negative charges. This helps to maintain the distinctive red colouration of the colloidal gold, but also to keep the particles stabilised within the colloid [115]. For BoNT detection, an assay has been developed that utilises this principle exploiting a visible colour change that occurs when colloidal gold is exposed to sodium chloride [116]. 

Figure 8 shows both the visual result of exposing gold colloid to salt both prior and after surface modification with a protective SNAP-25 protein layer [116]. The methodology developed utilises the process of BoNT cleaving SNAP-25; nine amino acids are removed from the C-terminus of the protein. This shortening of SNAP-25 decreases the protective layer present around the AuNPs in the gold colloid which in turn increase their susceptibility to aggregation upon exposure to the sodium chloride. The assay was able to detect BoNT/A via UV-visible spectroscopy on a semi-micro cuvette scale down to a detection level of 370 fg/mL, and even went on to further develop the assay on a microplate scale to increase the high throughput of samples, managing to retain a detection limit of 600 fg/mL [116]. Both of these methods are limited by the fact that SNAP-25 is exclusively cleaved by BoNT/A, C and E, with two of these contributing to cases of human botulism and the other to animal cases, the system would require further serotype testing with a different SNARE complex protein, which may result in multiplexing difficulties, which may limit its widespread use in the diagnostic and food sectors. It has a significant advantage with respect to its adoption in the pharmaceutical sector in that it is able to distinguish active toxin, which is a core require for MBA replacement in that area. It is also a very rapid methodology, taking ~10 min and costing around £1.40 per test [116]. The drawback of a minor decrease in sensitivity of the microplate assay is outweighed by it lowering both the time and cost required to 7 min and ~£0.90 per test. This detection method has real promise despite the drawbacks, and further exploration is needed to enable testing across all BoNT serotypes [116]. 

A similar methodology was developed utilising the same basic platform: BoNT cleaved peptides leading to AuNP agglomeration, facilitating a visually observable colour shift from red to blue [117]. The assay allowed for clear visually distinguishable responses at concentrations reaching as little as 1 ng/mL with a LoD around 250 pg/mL [117]. This method utilises each peptide fragment in a ‘turn-on’ method, where instead of the BoNT breaking a peptide linkage, destabilising the AuNPs (a ‘turn-off’ process), it uses Cu^2+^ to form a chemical complex with the post cleavage peptide fragments. This method is more efficient at inducing particle agglomeration which leads to the detection of lower concentrations of BoNT [117]. The main advantage of this methodology is simplicity of the assays coupled with the relatively high sensitivity; however, there are increased costs compared to the previously described colorimetric assay due to the use of AuNPs, Cu^2+^ and modified magnetic micro-particles [117]. It also, like most assays, has not been fully tested against various BoNT serotypes and in a range of sample matrices.

### 3.16. SERS Detection 

Surface-enhanced Ramon scattering (SERS) is a plasmonics readout technology that can be utilised in a point of care device. SERS consists of exploiting extrinsic Raman labels (ERLs) made up of gold nanoparticles that have been surface functionalised by Ramon reporter molecules (RRM), antibodies used for tracing (Abs) and a capture substrate, formed from antibodies bound to a gold surface that is cast onto a glass support [118]. The magnitude of the SERS signal quantifies the amount of antigen in a sample. There are several advantages that SERS has over ELISA and other molecular techniques. Firstly, the technique is able to differentiate at a higher level due to the tighter spectral properties observed in Ramon scattering as opposed to the electronic transitions utilised in optical absorbance and fluorescence. This is important when looking to develop a multi-marker pathogen platform. Secondly, an immunoassay using SERS is able to be analysed by a single excitation source which has implications for field deployment [118]. 

Lim et al. [118] reported on the detection of inactivated BoNT/A and BoNT/B using a sandwich immunoassay, gold nanoparticles and SERS in the context of two scenarios. The first mirrored the analysis required in response to a ‘white powder threat’, detecting the levels of BoNT/A and BoNT/B in phosphate-buffered saline, which is often used as a solvent of choice in recovering markers from powder-based dispersal matrices. The LoDs were 700 pg/mL and 84pg/mL for BoNT/A and B, respectively [118]. Scenario two was used to demonstrate that the assay could be used in a clinical situation by detecting the two neurotoxins in human serum. The LoDs were 700 pg/mL and 91 pg/mL, again in respect to BoNT/A and B [118]. In both instances, the time to result was under 23 h. These LoDs show that this technique can detect low levels of BoNTs in two appropriate matrices, but this lack of complex sample matrices is seen as a disadvantage in the diagnostic and food testing fields as well and the non-multiplex nature of detectable serotypes. It is also unable to distinguish toxin potency that is quantity of active toxin which will be a major barrier to this method being adopted by the pharmaceutical sector. These results, together with recent advances in the development of mobile Ramen instruments, suggest that this technique could be used in the field, if the issue of sample matrices can be addressed and a more simultaneous system for serotype detection can be configured. Further work is now being undertaken to simplify the process and speed up detection time (<1 h), including the use of a handheld Ramen spectrometer for portable readouts [118].

### 3.17. Cell-Based Assays

Cell-based assays have the ability to replicate some of the aspects of botulism seen in vivo. Several assays have been designed to use continuous cell lines or primary neurons obtained from bird or rodent spinal cord cells [50,119]. Figure 9 shows a simplistic overview of a typical cell-based platform utilised for the detection of BoNT. Continuous cells line-based assays have not demonstrated great sensitivity in detecting BoNT in complex matrices, whereas a sensitivity comparable to the MBA has been seen with some primary cell line assays [50]. 

As a model for BoNT detection, cell-based assays can be utilised in several areas of interest: cell surface receptor binding, endocytosis internalisation, membrane translocation and SNARE cleavage, this combination makes cell-based assays a prime candidate for BoNT inhibitor screens [6,44]. For their use in the food or diagnostic sectors, they do have several disadvantages, speed (days for detection), it is technically demanding and requires facilities and the expertise to maintain cell lines, sensitivity is in the range 1–10 ng/mL [6,50,120].

Recently Rust et al. proposed a SiMa cell-based assay approach where, when used in combination with re-engineered VAMP molecules, BoNT/B detection achieved sensitivity comparable to the MBA [39]. It combines the cell-based approach with ELISA in a one-step assay that facilitates detection of BoNT/B via a luminescent enzymatic reaction. The limit of detection is ~3 pg/mL with an analysis time similar to that of ELISA, although the preparation of the cell reporter line prior to analysis is more time consuming [39]. This method’s ability to distinguish active toxin is of high importance to the pharmaceutical industry, where speed and cost are less of a concern. This detection method presents as advantageous due to the user-friendly nature of the assay and its alignment with the 3Rs concept; which describes the steps being taken to replace, refine and reduce animal research to replace the need for animal use. The cell reporter line could also be further adapted to enable its use in real-time during the manufacture of botulinum vaccines [39]. 

## 4. Conclusions

There are many different detection platforms that have been developed for their ability to be utilised in the detection of botulinum neurotoxins. They all have a common goal of replacing the MBA and all of which are trying to address the 3Rs; replacement, refinement and reduction of animal research and use [42]. The ability for a detection method to detect active toxin levels is of particular interest to the pharmaceutical production industry. A methodologies ability to be able to detect where possible multiple BoNT serotypes, as well as the samples being comprised of complex matrices is of particular interest in the diagnostic and food testing sectors. Several of the proposed detection methods have sensitivity levels that are lower than that of the MBA, with some able to also distinguish toxicity of the BoNT with the ability to detect active toxin levels (highlighted in Table 3), and all of the detection methods have a faster response times for analysis of samples. This review also pertained to examine the costs of the detection methods, the estimated costs per test have been estimated based on the information available from the detection methods published work. 

In addition to the methods listed in Table 3, there are various other techniques that have seen development for their use as BoNT detection platforms. One of these such systems is surface plasmon resonance, this has seen development from initial uses [121,122] to more recent and emerging advancements [81,123]. Other such techniques that were first explored several years ago are time-resolved fluorescence assay [124] and liquid chromatography [125]; these have more recently been explored, providing a relatively low level of detection and short detection time [40,126]. The problem with these techniques as detection platforms, is that they often require expensive instrumentation or have simply not been tested across a wide enough set of parameters. These factors have led to other detection methods stagnating and not progressing in advancement such as capillary electrophoresis [127] and even a micro-mechanical sensor able to detect synaptobrevin molecule cleavage [128]. 

For the detection methods summarised in this review, all have an estimated cost per test that is lower than the cost of the mouse bioassay. However, these costings do not factor in instrumentation laboratory/facilities or staff overheads. For some of the detection methods despite the relatively low-test costs the additional expenses hinder their viability. To truly replace the mouse bioassay, detection methods will not only need to be able to provide similar levels of sensitivity to BoNT toxin, but do so at a lower cost while being easy to use. However, the function with which they are used will require key parameters to be achieved. For adoption in pharmaceutical product testing, the methods will need to also be able to distinguish and quantify active toxin levels. In contrast, for diagnostic and food testing, this is less of an issue, and instead adoption in these sectors would require the detection method to have the ability to detect BoNT in complex samples matrices such as serum, stool, culture supernatants and a diverse set of food samples. It would also provide a huge advantage if the assay were able to easily detect multiple BoNT serotypes from one sample simultaneously.

Based on the three principles that we have looked at—sensitivity, speed and cost—there are four detection methods that present as the most promising as replacements to the MBA utilising the principles of the 3Rs. The four platforms can be seen in Figure 10 and are located in the middle of the Venn diagram where all of the desired characteristics converge. These are the centrifugal microfluidic assay [90], colorimetric assay [116], SPR-based biosensor [81] and the EIS platform [101]. When summarising the advantages and disadvantages of these different methods we concluded that for the centrifugal microfluidic assay the low cost, ease of use, the ability to use a wide base of sample matrices and the speed and sensitivity were clear advantages in the use of this detection method. The main disadvantages in the use of this methodology is both the need to perform additional analysis to obtain the final quantitative result, and the costs associated with both the additional analysis method and the expected high cost of the assay device, which is still at the prototype stage [90]. With regard to the colorimetric assay, the very low cost per test, the rapid detection time from sample to result and the high levels of sensitivity provide considerable advantage to this detection platform. The main limitation is that it requires lab-based equipment and analysis which limits its wider scale usage and increases the cost. This can be somewhat lessened by the fact that this methodology should be easily adaptable to a more portable field device [129]. SPR-based biosensors and the EIS platform both provide similar advantages in their use for the detection of BoNT. Both are cheap, quick and provide greater sensitivity than the MBA; however, they are limited again by their constraint to a laboratory-based setting due to the need for instrumentation and result analysis. Another main drawback is their limited testing of complex sample matrices [81,101]. It is hoped that these methodologies can be scaled down and be made more portable utilising advances from other areas of biosensors and diagnostics.

In summary, the main goal in the field of botulinum neurotoxin detection is the replacement, refinement and reduction of animal usage regardless of the sector in which the testing is required. This means a move away from the use of the mouse bioassay, the current gold standard in BoNT detection, which uses an estimated 600,000 mice each year worldwide [42]. To do this, the alternative will not only need to present increased sensitivity, rapid sample analysis and low cost, in the pharmaceutical production sector, the ability to quantify active toxin levels is paramount with speed of detection less of a priority. In the diagnostic and food testing sectors, the ability to detect multiple serotypes simultaneously from complex sample matrices is the key requirement, while retaining the need for rapid low-cost detection. This review has looked at detection methods that are in development and identified the most promising technologies. These will all need optimisation moving forward to overcome the remaining hurdles preventing them from wide scale multi-industry usage. The utilisation of interdisciplinary advances in the fields of chemistry, biology and engineering will hopefully facilitate a commercial alternative that conforms to the 3Rs to finally replace the MBA. 

## Figures and Tables

**Figure 1 toxins-11-00418-f001:**
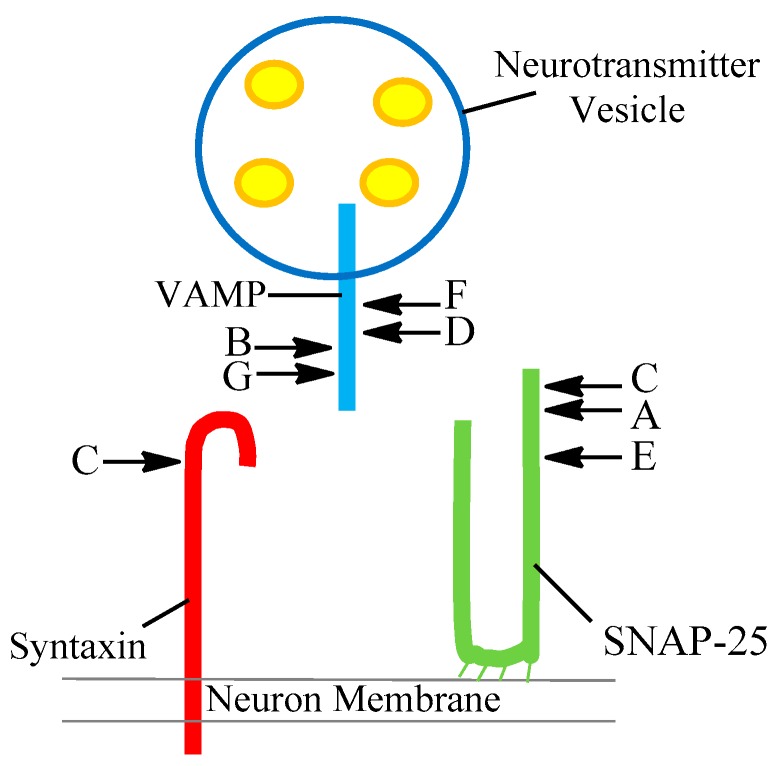
Diagram showing the proteins that make up the Soluble N-ethylmaleimide-sensitive factor activating protein receptor (SNARE) complex; Syntaxin (Red), SNAP-25 (Green) and Synaptobrevin, also known as vesicle-associated membrane protein (VAMP) (Blue). Additionally, botulinum neurotoxin serotype is listed next to the corresponding protein it is responsible for fragmenting, along with the location on the protein at which it cleaves.

**Figure 2 toxins-11-00418-f002:**
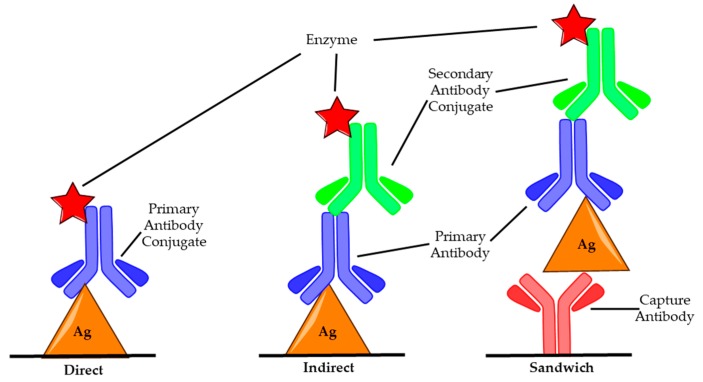
Image of common ELISA set ups: Direct, Indirect and Sandwich. In both direct and indirect ELISA, antigens (Ag) are bound to the microtiter plate first, an antibody specific to the antigen is then introduced. In direct assays this primary antibody (blue) has been modified with an enzyme (red star) such as HRP, which when exposed to a substrate produces a measurable colour change. In indirect ELISA, this enzyme is bound to a secondary antibody (green) that has been modified with an enzyme to facilitate colour change; this secondary antibody binds to the primary antibody. In sandwich ELISA, the surface is treated with a capture antibody (red) specific to a desired antigen before the rest of the assay proceeds in the same manner as the indirect assay.

**Figure 3 toxins-11-00418-f003:**
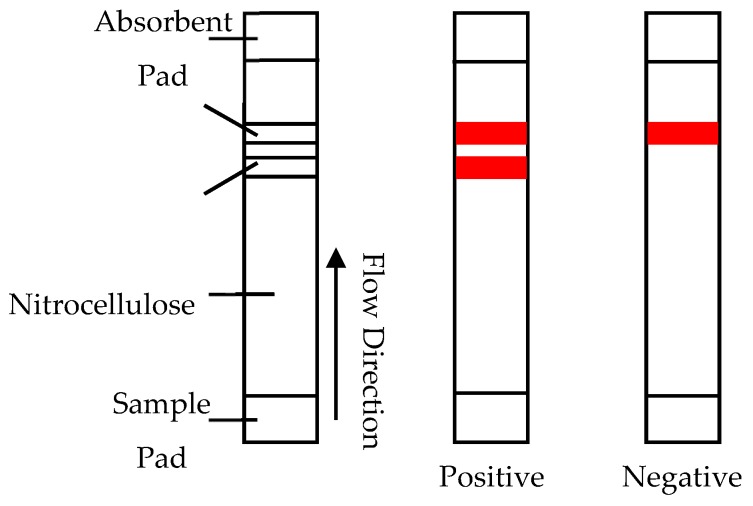
Diagram of lateral flow assay and examples of positive and negative results [67].

**Figure 4 toxins-11-00418-f004:**
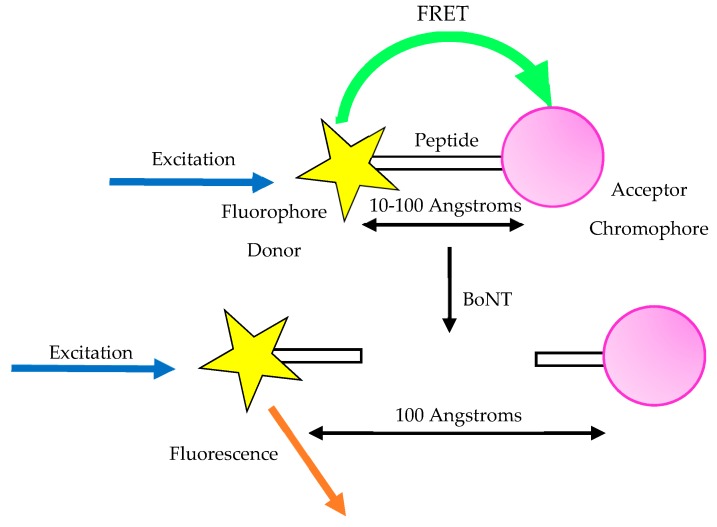
Schematic explaining the action of the FRET assay. Typically, an acceptor chromophore (pink) is linked to a fluorophore donor (yellow) via a peptide; this allows the transfer of energy resulting in FRET being detected. Upon exposure to botulinum neurotoxin, the peptide linker is cleaved and fragmented allowing the chromophore and fluorophore to dissociate. This inhibits the transfer of energy in the system.

**Figure 5 toxins-11-00418-f005:**
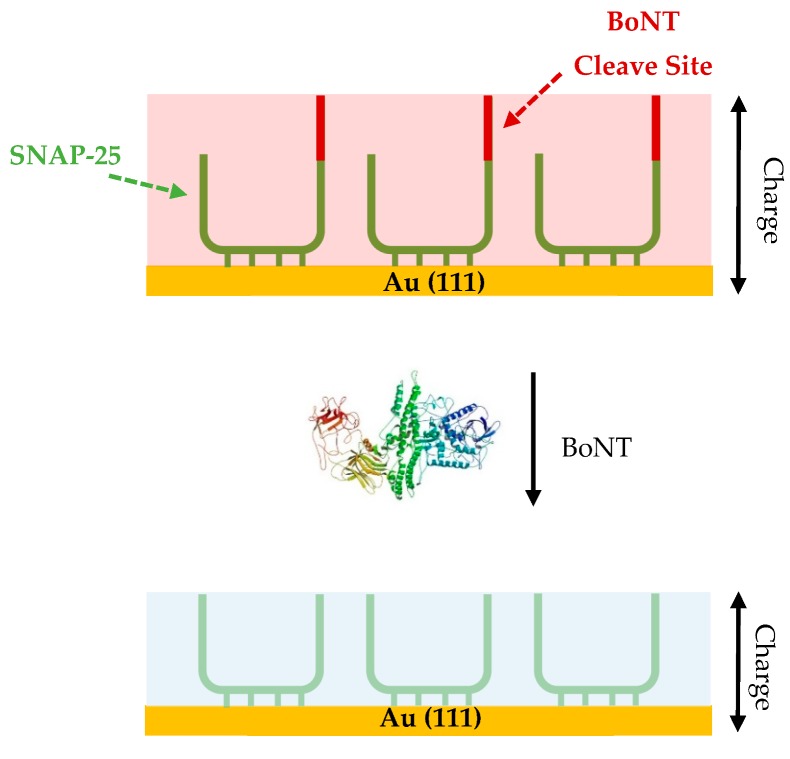
Schematic showing how botulinum neurotoxin is detected by measuring the decrease in anodic peak charge observed when BoNT cleaves a section of the self-assembled SNAP-25 monolayer.

**Figure 6 toxins-11-00418-f006:**
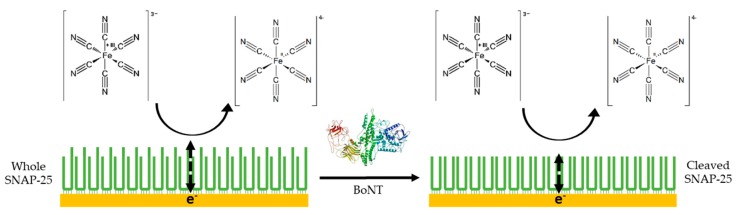
Schematic showing how whole and cleaved SNAP-25 differs in blocking the reduction of the redox probe, with the whole SNAP-25 providing a greater blocking ability of the redox probe due to its larger size.

**Figure 7 toxins-11-00418-f007:**
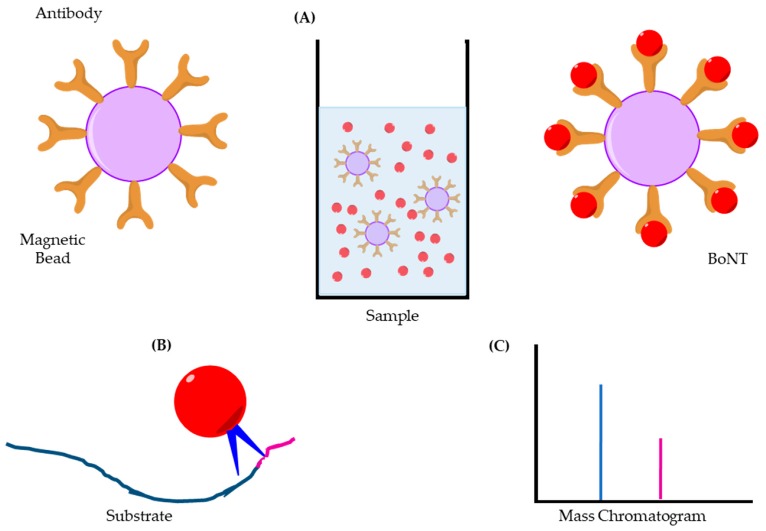
An overview of the endopeptidase mass spectrometry assay, serotype specific antibodies are conjugated to magnetic beads then added to a sample (**A**); the beads are removed and thoroughly washed before a substrate that mimics the toxins natural target is added (**B**). The solution is then incubated, and the resulting mixture is analysed by mass spectrophotometry. Both whole substrate and cleaved fragments that result from incubation with BoNT can be detected (**C**).

**Figure 8 toxins-11-00418-f008:**
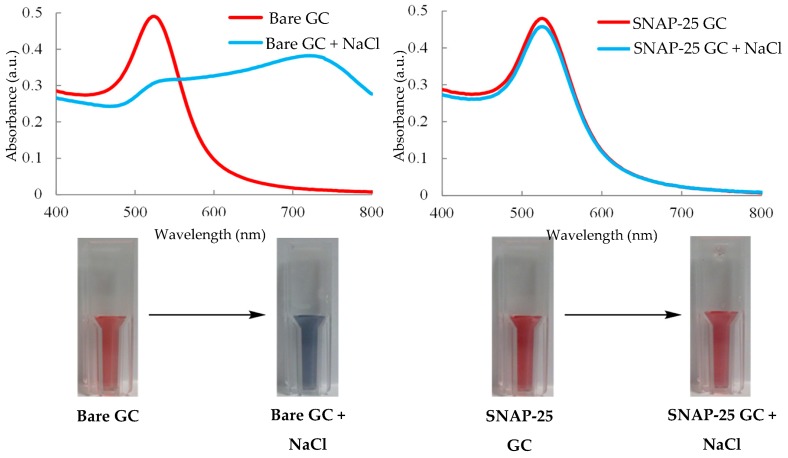
UV-visible spectra showing un-modified (**left**) and modified (**right**) gold colloids (GC), detailing that upon addition of sodium chloride, the unmodified colloid decreases in stability and undergoes aggregation. The inset images show the visible colour shift [116].

**Figure 9 toxins-11-00418-f009:**
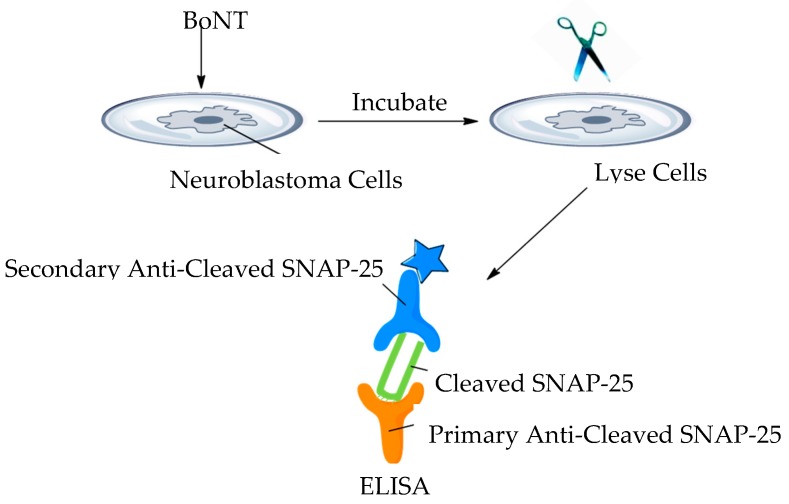
Schematic of cell-based assay for botulinum neurotoxin.

**Figure 10 toxins-11-00418-f010:**
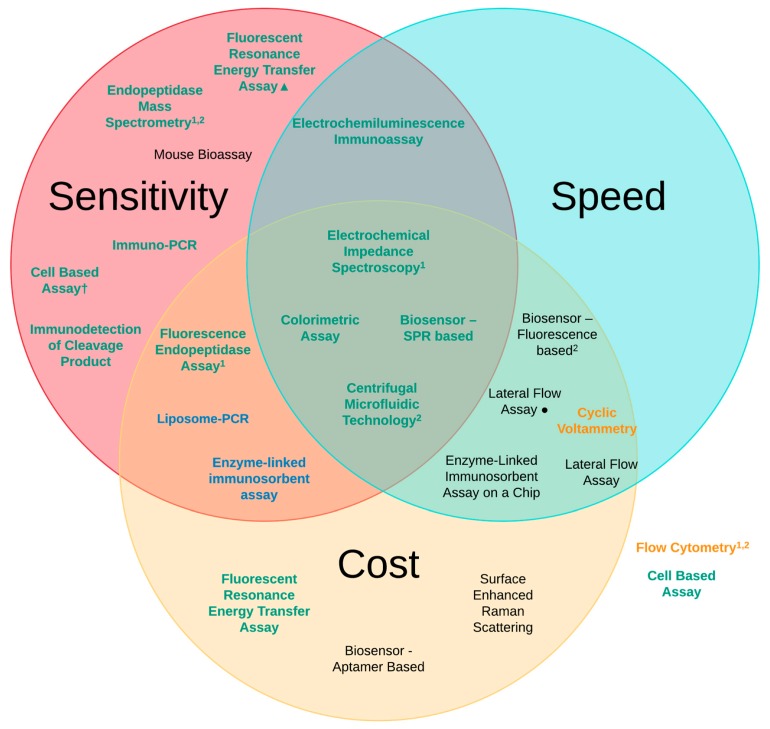
A visual representation of the BoNT detection methods analysed in this review. The assays were compared using the mouse bioassay sensitivity as the baseline (10 pg/mL), an analysis time of <1 h and cost of <£30 per test. Highlighted are a method’s ability to distinguish active toxin and improved sensitivity over MBA (Green), active toxin quantification but poorer LoD than the MBA (Orange) and no active toxin quantification by still-improved sensitivity over MBA (Blue). The two detection methods that fall outside of satisfying the parameters of speed, cost and sensitivity are placed outside of the Venn diagram. Additionally, the following notation are used: (●) combined with endopeptidase activity assay, (▲) with an additional immunoseparation step, (†) cell reporter line coupled with ELISA detection, (1) have been tested in multiplex systems/with multiple serotypes and (2) complex sample matrices tested.

**Table 1 toxins-11-00418-t001:** Bacteria-producing serotypes of botulinum neurotoxin.

Bacteria	Group	Serotype	Human Botulism
*C. Botulinum*	I	A,B,F	Yes
	II	E,B,F	Yes
	III	C,D	No
*C. Argentinense*	IV	G	No
*C. Baratii*	V	F	Yes
*C. Butyricum*	VI	E	Yes

**Table 2 toxins-11-00418-t002:** The target substrate and cleavage site of each serotype of the toxin.

Serotype	SNARE Protein	Cleavage Site
A	SNAP-25	197–198
B	Synaptobrevin (VAMP)	76–77
C	SNAP-25 Syntaxin	198–199 253–254
D	Synaptobrevin (VAMP)	59–60
E	SNAP-25	180–181
F	Synaptobrevin (VAMP)	58–59
G	Synaptobrevin (VAMP)	81–82
FA	Synaptobrevin (VAMP)	54–55

**Table 3 toxins-11-00418-t003:** Summary table detailing the detection methods for BoNT. The assays that have lower limits of detection than the mouse bioassay along with the ability to detect active toxin are highlighted in green. Those that are just more sensitive than the MBA, but with no distinction for active toxin, are highlighted in blue. Finally, methods that can distinguish active toxin but have LoDs greater than the MBA are highlighted in orange.

Detection Method	Sensitivity of Analysis (LoD)	Speed of Analysis	Estimated Cost (Per Test)
Mouse Bioassay (MBA) [6,41]	10 pg/mL	4–6 days	£320
Enzyme-linked immunosorbent assay (ELISA) [6,41]	2 pg/mL	6 h	£25
Lateral Flow Assay (LFA) [67,70]	50 pg/mL 20 pg/mL^●^	~10 min ~12 h	£3 £25
Immuno-PCR [72]	1 pg/mL	6–9 h	£45
Liposome-PCR [73]	20 ag/mL	7–9 h	£25
Enzyme-Linked Immunosorbent Assay on a Chip (EOC) [75]	2 ng/mL	~30 min	£20
Biosensor—Fluorescence-based [77]	150 pg/mL	~10 min	£20
Biosensor—Aptamer-based [80]	40 pg/mL	~24 h	£12
Biosensor—SPR-based [81]	6.76 pg/mL	~20 min	£28
Fluorescent Resonance Energy Transfer Assay (FRET) [6,83]	60 pg/mL 1 fg/mL^▲^	3 h 2 h	£11 £50
Flow Cytometry [86]	50 pg/mL	~4 h	£40
Fluorescence Endopeptidase Assay [6,88]	3 pg/mL	3 h	£20
Centrifugal Microfluidic Technology [90]	90 fg/mL	~30 min	£2
Electrochemiluminescence Immunoassay (ECL) [92]	5 pg/mL	~40 min	£50
Cyclic Voltammetry (CV) [94]	250 pg/mL	~15 min	£20
Electrochemical Impedance Spectroscopy (EIS) [101]	25 fg/mL	~35 min	£24
Immuno detection of Cleavage Product [105]	440 fg/mL	6 h	£40
Endopeptidase Mass Spectrometry [109]	100 fg/mL	4–8 h	£40
Colorimetric Assay [116]	370 fg/mL (cuvette) 600 fg/mL (96-well plate)	~10 min ~7 min	£2 £1
Surface-enhanced Ramon scattering (SERS) [118]	84-700 pg/mL	~23 h	£25
Cell Based Assays [6,39,50,120]	1–10 ng/mL 3 pg/mL^†^	2–3 days	£35 £45

● Combined with endopeptidase activity assay ▲ With additional immunoseparation step, † Cell reporter line coupled with ELISA detection.

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
