# Peer review of "Rapid Detection of Botulinum Neurotoxins—A Review"

_toxins, 2019, doi:10.3390/toxins11070418_

Round 1
Reviewer 1 Report
This manuscript is well written in detail about detection of Botulinum neurotoxins. In addition, the point of analysis in terms of sensitivity, speed, and cost regarding the detection of toxins is excellent. This manuscript is judged to be suitable for Toxins.
Author Response
Thanks for the positive review
Reviewer 2 Report
The authors give a brief survey of the biochemistry and the pathology of botulinum intoxication and discuss in detail the different methods of qualitative as well as quantitative detection methods for botulinum toxins in different sample matrices.
The manuscript gives a well-balanced introduction into the principles of suitable assay systems, their uses and their limitations. Assays relying on direct detection of toxin molecules are described, and indirect assays detecting potential proteolytic products of the botulinum toxin are considered as well. All methods are thoroughly compared according to the three criteria sensitivity, speed and costs.
The text is well written and provides a very valuable introduction to the topic for generally interested readers. For people involved in routine testing of biological or food samples this review provides valuable assistance for making decisions about suitable analysis methods, especially those avoiding the use of live animals.
Minor points:
l. 27: To avoid misunderstanding, I suggest to use the term 'proprotein' instead of 'protein' in this sentence.
l. 188: maybe better: "BoNT works by suppression of transmitter release from endings of motorneurons, resulting in flaccid paralysis ..."
l. 189: Is reference [6] the most appropriate citation here? I would suggest to use a citation of a publication that describes the mechanistic details of toxin effects, e.g. Ref. [37]. The optimal fit of citations to the information in the text should be checked throughout the manuscript as I feel that there are other cases like the one described above.
l. 204: maybe better: "The quick onset of the disease, the extreme toxicity of BoNTs, and the absence of treatments to reverse paralysis [...], means that a quick BoNT detection method is required and that it has to be both sensitive and specific."
Author Response
We have read the review reports and have made the following changes (all of which have been tracked in Microsoft Word) to the manuscript as suggested by the reviewers:
l. 27: To avoid misunderstanding, I suggest to use the term 'proprotein' instead of 'protein' in this sentence.
Wording has been changed as suggested.
l. 188: maybe better: "BoNT works by suppression of transmitter release from endings of motor neurons, resulting in flaccid paralysis ..."
Wording has been changed, should now read as: “BoNT works via suppression of transmitter release from endings of motor neurons, resulting in flaccid paralysis.”
l. 189: Is reference [6] the most appropriate citation here? I would suggest to use a citation of a publication that describes the mechanistic details of toxin effects, e.g. Ref. [37]. The optimal fit of citations to the information in the text should be checked throughout the manuscript as I feel that there are other cases like the one described above.
Reference [6] has been changed to reference [37] in this case as suggested, and having looked over the manuscript the following additional reference changes have been implemented:
Line 405: ref [6] changed to ref [82] | Line 499: ref [6] changed to ref [91] |
Line 418: ref [6] expanded to ref [6,82,83] | Line 502: ref [6] changed to ref [6,92] |
Line 425: ref [6] changed to ref [83] | Line 506: ref [6] changed to ref [6,92] |
Line 428: ref [6,41] changed to ref [6,41,84] | Line 582: ref [6] changed to ref [6, 105] |
Line 442: ref [6] changed to ref [85,86] | Line 729: ref [6] changed to ref [6,44] |
Line 444: ref: [41] changed to ref [85] | Line 731: ref [6,41] changed to ref [6,50,120] |
Line 470: ref [6] changed to ref [88] |
l. 204: maybe better: "The quick onset of the disease, the extreme toxicity of BoNTs, and the absence of treatments to reverse paralysis [...], means that a quick BoNT detection method is required and that it has to be both sensitive and specific."
Wording has been changed, should now read as: ‘The quick onset of the disease, the extreme toxicity of BoNTs and the absence of treatments to reverse paralysis […], means that a quick BoNT detection method is required and that it is both sensitive and specific.’